# Comparative Microbial Profiles of Colonic Digesta between Ningxiang Pig and Large White Pig

**DOI:** 10.3390/ani11071862

**Published:** 2021-06-23

**Authors:** Linfeng Lei, Zhaobin Wang, Jianzhong Li, Huansheng Yang, Yulong Yin, Bie Tan, Jiashun Chen

**Affiliations:** 1Animal Nutritional Genome and Germplasm Innovation Research Center, College of Animal Science and Technology, Hunan Agricultural University, Changsha 410125, China; chenjs1988@126.com (L.L.); yinyulong@isa.ac.cn (Y.Y.); bietan0412@hotmail.com (B.T.); 2Hunan International Joint Laboratory of Animal Intestinal Ecology and Health, Laboratory of Animal Nutrition and Human Health, College of Life Sciences, Hunan Normal University, Changsha 410081, China; anywang0914@yahoo.com (Z.W.); yhs@hunnu.edu.cn (H.Y.); 3Hunan Provincial Key Laboratory of Animal Nutritional Physiology and Metabolic Process, Scientific Observing and Experimental Station of Animal Nutrition and Feed Science in South-Central, Ministry of Agriculture, Hunan Provincial Engineering Research Center for Healthy Livestock and Poultry Production, Key Laboratory of Agro-Ecological Processes in Subtropical Region, Institute of Subtropical Agriculture, Chinese Academy of Sciences, Changsha 410125, China

**Keywords:** pig breeds, colon, short chain fatty acids, intramuscular fat, gut microbiome

## Abstract

**Simple Summary:**

Different breeds of pigs vary greatly in their propensity for adiposity. The gut microbiome plays a crucial role in shaping host physiological responses. However, it remains unclear how the gut microbiota influences host growth, in particular adipogenesis. This study aimed to compare microbial profiles in the colons of two pig breeds.

**Abstract:**

Sixteen 35-day-old piglets, including eight Large White (LW) piglets (a lean-type pig breed) and eight Ningxiang (NX) piglets (a fatty-type Chinese Indigenous pig breed), were fed the same diet for 105 days. NX pigs had higher intramuscular fat content than LW pigs (*p* < 0.05). According to 16S rRNA gene sequencing, the relative abundances of the genera Ruminococcaceae*_NK4A214_*group, *Parabacteroides,* Christensenellaaceae*_R-7_*group and *Ruminiclostridium* were higher, whereas the abundances of Prevotellaceae_*NK3B31_*group, *Prevotella*, *Subdoligranulum* and *Faecalibacterium* were lower, in the colon of NX pigs compared to that of LW pigs. Nonmetric multidimensional scaling analysis revealed that the microbiota of the two pig breeds clustered separately along the principal coordinate axis. Furthermore, functional prediction of the bacterial communities suggested higher fatty acid biosynthesis in NX pigs. NX pigs also exhibited lower concentrations of total short-chain fatty acids, propionate and butyrate in the colon (*p* < 0.05). These findings suggest that NX pigs exhibited higher intramuscular fat content and backfat thickness than LW pigs. The bacterial communities in the colon of NX pigs were also more diverse than those in the colon of LW pigs, which might be used as a potential metabolomics mechanism to research different breeds of pigs.

## 1. Introduction

Global pork production has made significant contributions to food security. Pork is the most commonly produced meat and accounts for approximately 40% of all meat consumed worldwide [1]. Animal fat deposition is a complex biological process. Fat deposited in muscle includes intramuscular fat (IMF) and intermuscular fat, and IMF is a key factor affecting meat qualities, such as tenderness, juiciness and taste [2], making it an economically important factor in pig breeding. However, abnormal or excessive accumulation of fat in the body can lead to obesity, which is harmful for animal health [3].

The intestine is the major site of food transformation and metabolism, and the gut microbiota is important for nutrient and energy metabolism and for maintaining the homeostasis of the host immune system [4]. It is also influenced by many factors, such as genetics, disease, diet, environment and lifestyle [5]. It has been shown that the gut microbial profile is highly related to obesity [6]. Imbalances in the proportion of various components of the intestinal microbiota contribute to the development of obesity via several mechanisms, including the storage of nutrients and energy, elevated levels of systemic inflammation and increased lipid deposition [7,8]. In the gastrointestinal tract, the highest microbial content is found in the colon. Here, fermentation of undigested carbohydrates, proteins and enzymes by gut microbes results in the production of short-chain fatty acids (SCFAs), ammonia and biogenic amines. Some metabolites can be used as energy sources by the host, whereas others affect host metabolic functioning and health [9]. Some metabolites have even been shown to regulate the host brain, which can also affect obesity [10].

Previous studies have confirmed differences in the serum and cecal metabolomic profiles between two pig breeds [11,12], as well as differences in colonic bacterial abundances and bacterial metabolites between fatty- and lean-type pigs [13]. Furthermore, the gut microbiome is shaped by host diet and host genotype and can affect the postnatal development of gut tissues and host metabolic health [14]. Recent studies have demonstrated that the gut microbiota plays a pivotal role in contributing to adiposity in different breeds of pigs [15].

Ningxiang (NX) pigs, a well-known Chinese Indigenous fatty-type breed, exhibits high IMF content, early sexual maturity and low growth rates [16]. In contrast, the Large White (LW) pig, also known as the English Large White, is a lean-type pig breed characterized by a fast growth rate and high lean meat content [17]. It has been reported that the influence of genetic factors leads to differences in the sensory quality of pork [18]. Thus, these two porcine breeds are suitable models to investigate the genetic differences and molecular mechanisms underlying the above-mentioned phenotypic differences [19]. However, there is little information regarding the mechanisms that control these phenotypic differences or the effects of microbial differences on host phenotypes. To understand the relationship between swine growth, adipogenesis and gut microbiota, we used 16S rRNA gene sequencing to analyze and identify the microbiota involved in regulating differences in adipogenesis between NX and LW pigs. This study will help develop a comprehensive understanding of the mechanisms by which the gut microbiota regulates fat deposition in these two pig breeds.

## 2. Materials and Methods

Experiments involving the use of pigs were subjected to an approval process under national guidelines by Chinese guidelines for animal welfare and experimental protocols, and all procedures were approved by the Hunan Agricultural University Animal Care and Use Committee (Changsha, China) (permit number: CACAHU 2021-00116).

### 2.1. Animals, Housing and Experimental Design

Eight LW barrows (10.90 ± 0.36 kg body weight (BW)) and eight NX barrows (8.18 ± 0.52 kg BW) weaned at 35 days of age were fed the same basal diet based on the Nutrient Requirements of Livestock and Feeding Standards of Pig (NY/T 65-2004). The pigs were housed individually in cages (1.5 × 1.5 m^2^) equipped with a double-sided feeder and a stainless steel nipple drinker. All pigs were given ad libitum access to feed and clean drinking water throughout the 105-day experiment. The feed intake of each pig was recorded daily, and each pig was weighed at the beginning and the end of trial to calculate the average daily gain (ADG), the average daily feed intake (ADFI) and the feed:gain ratio (F/G).

### 2.2. Sample Collection and Preparation

At the end of the trial, all pigs were fasted for 24 h and transported to the slaughterhouse for slaughter by bleeding after electrical stunning. Samples from the longissimus dorsi (LD), soleus (SM) and quadriceps (QF) muscles were placed in vacuum bags at 4 °C to measure meat quality traits. In addition, samples from the LD, SM and QF muscles were immediately frozen in liquid nitrogen and stored at −80 °C for further analysis. After chilling for 24 h, muscle samples were collected and stored at −80 °C for protein solubility measurements. The contents of the colon were collected, immediately frozen in liquid nitrogen and stored at −80 °C until 16S rRNA and SCFAs content were assessed.

### 2.3. Carcass and IMF

After evisceration, the right half of the carcass was weighed and used to assess morphometric parameters. Carcass quality was evaluated based on final weight, carcass weight, dressing percentage, fat percentage, lean percentage and average backfat thickness, as previously described [20]. Drip loss was defined as the weight loss of a meat sample (50 g) placed on a flat plastic grid, wrapped in foil and stored for 24 h (24 to 48 h postmortem) in a refrigerator at 4 °C [18]. The IMF content of the muscle was analyzed in duplicate using a Soxhlet Extractor (SER-148, Italy VELP Co., Usmate Velate, Italy) with petroleum ether (boiling temperature range: 60–90 °C).

### 2.4. DNA Extraction, 16S rRNA Sequencing and Bioinformatics Analysis

Total bacterial genomic DNA was extracted from the colonic digesta of each pig using a QIAamp DNA Stool Mini Kit (Qiagen, Hilden, Germany) according to the manufacturer’s instructions. DNA quality was evaluated by 1% agarose gel electrophoresis. The V3-V4 hypervariable regions (338F: ACTCCTACGGGAGGCAGCAG, 806R: GGACTACHVGGGTWTCTAAT) of the 16S rRNA gene (~460 bp) were amplified by polymerase chain reaction (PCR) from microbial genomic DNA using a GeneAmp 9700 thermocycler (Applied Biosystems, Foster City, CA, USA) according to a previous study [21]. PCR products were purified using a GeneJETTM Gel Extraction Kit (Thermo Scientific, Waltham, MA, USA). Amplicons from different samples were mixed in equal amounts. Sequencing libraries were generated using an Ion Plus Fragment Library Kit 48 rxns (Thermo Scientific) following the manufacturer’s recommendations. Purified amplicons were pooled in equimolar amounts and paired-end sequenced (2 × 250 bp) on an Illumina MiSeq 2500 platform (Novogene, Beijing, China).

Quality filtering of the raw reads was performed under specific filtering conditions to obtain high quality clean reads (Q > 25) using QIIME software (version 1.9.1) (http://cutadapt.readthedocs.io/en/stable/, accessed on 2 December 2015) [22]. The reads were compared with the reference database (Gold Database, http://drive5.com/uchime/uchime_download.html, accessed on 15 August 2011) using the UCHIME algorithm (http://www.drive5.com/usearch/manual/uchime_algo.html, accessed on 15 August 2011) [23] to detect chimera sequences, which were removed. All clean reads of all samples were clustered using UPARSE software (version V7.0.1001; http://drive5.com/uparse/, accessed on 18 August 2013), and the sequences were clustered into operational taxonomic units (OTUs) with a 97% similarity level cutoff. For each representative sequence, the Silva Database (https://www.arb silva.de/, accessed on 27 November 2012) [24] was used based on the Mothur algorithm to annotate taxonomic information. Alpha diversity (observed species, Chao1, Shannon, Simpson, abundance-based coverage estimator (ACE) and Good’s coverage) was calculated using QIIME software (version 1.7.0) and displayed with R software (version 2.15.3) (https://www.r-project.org/) (R Core Team, Vienna, Austria). Nonmetric multidimensional scaling (NMDS) plots based on Bray–Curtis dissimilarities were created to show group differences in microbial community structure. R statistics were calculated using analysis of similarities (ANOSIM), with an R value near +1 indicating that there was dissimilarity between the groups and an R value near 0 indicating no significant dissimilarity between the groups. Samples were considered to be significantly different at *p* < 0.05. Venn graphs were created using Venn diagram software (https://bioinformatics.psb.ugent.be/webtools/Venn/, accessed on 15 August 2011). Raw data were deposited into the SRA database with accession number PRJNA672239.

### 2.5. Metagenomic Functional Predictions

Predictive functional profiling of microbial communities was conducted using PICRUSt (phylogenetic investigation of communities by reconstruction ofunobserved states). Based on the Kyoto Encyclopedia of Genes and Genomes (KEGG) database, PICRUSt can predict the principal functions of the corresponding 16S rRNA OTUs. OTUs were selected against the Greengenes version 13.5 database according to instructions provided by the Genome Prediction Tutorial for PICRUSt. The output files from the PICRUSt analysis were uploaded to Metagenomic Profiles version 2.1.3 software for further statistical analysis and graphical depiction of all predictive functional data sets.

### 2.6. Analysis of SCFAs

To ensure consistency, chyme samples were dried at low temperature using a vacuum freeze dryer (FreeZone 2.5, Labconco, Kansas City, MO, USA). After the dried samples were pulverized and mixed, 0.5 g was moved to a 10 mL centrifuge tube, mixed with 5 mL double distilled water for 30 min and centrifuged at 15,000 rpm for 10 min at 4 °C. The supernatant was taken, followed by the mixing of the solution twice and adjustment of the final volume to 10 mL. The obtained liquid (0.9 mL) was mixed with 0.1 mL 25% metaphosphoric acid solution in a 1.5 mL centrifuge tube and stored in a refrigerator at 4 °C for 3 h. The mixture was then centrifuged at 15,000 rpm for 10 min at 4 °C, and the supernatant was filtered through a 0.22 microporous membrane and analyzed using an Agilent 7890 gas chromatograph (Agilent Technologies, Inc., Palo Alto, CA, USA) as previously described [25].

### 2.7. Statistical Analysis

Growth performance, SCFAs content and the relative abundances of bacterial phyla and genera were analyzed using Student’s *t*-tests. All data were analyzed using SPSS Statistics for Windows, Version 20.0 (IBM Corp., Armonk, NY, USA). Data are presented as means ± standard error of the mean (SEM), and differences were considered significant at *p* < 0.05.

## 3. Results

### 3.1. Growth Performance and Carcass Traits

Growth performance and carcass traits were compared between NX and LW pigs at 105 days of age as shown in Table 1. NX pigs exhibited higher backfat thickness than LW pigs (*p* < 0.05), but ADG, carcass weight and drip loss were lower in NX pigs (*p* < 0.05).

### 3.2. IMF Content

The IMF content of the two pig breeds is listed in Figure 1. NX pigs had a higher IMF content in the LD and QF muscles than LW pigs (*p* < 0.05). There were no significant differences in IMF content in the SM muscle between the two breeds (*p* > 0.05).

### 3.3. Microbiome Sequencing

To assess differences in the intestinal microbiota composition of NX and LW pigs, 16S rRNA sequencing was performed using bacterial DNA isolated from the colonic digesta of 105-day-old pigs. After filtering, 1.13 million clean reads were produced (73,705 ± 2205 reads per sample) (Appendix A). In order to study the species composition and diversity of the samples, the clean read sequences from all samples were clustered into OTUs with 97% identity. A total of 1119 and 1085 OTUs were detected in NX and LW pigs, respectively, including 981 common OTUs (Figure 2A). The Good’s coverage index was approximately 99.6%, indicating that sufficient depth of sequencing and adequate data were achieved.

### 3.4. Bacterial Composition and Diversity

At the phylum level, the taxonomic classification of the clustered OTUs of the colonic microbiota revealed the presence of 10 bacterial phyla. Firmicutes were the most predominant, representing 47.36–74.27% of the bacterial population of both NX and LW pigs. Bacteroidetes constituted the second most abundant phylum, representing 20.41–36.64%, followed by Spirochaetes (0.75–27.11%) and Proteobacteria (0.87–1.94%). The proportion of other phyla (Fusobacteria, Tenericutes, Cyanobacteria, Actinobacteria, Verrucomicrobia and Saccharibacteria) was less than 1% of the total microbial community. Widespread differences were found in the gut microbial community structure of NX pigs compared to that of LW pigs. In particular, in NX pigs, abundances of Firmicutes (64.89% vs. 61.48%) (*p* = 0.449), Proteobacteria (4.95% vs. 2.48%) (*p* = 0.292) and Fusobacteria (64.89% vs. 61.48%) (*p* = 0.449) were higher, but abundances of Bacteroidetes (25.42% vs. 27.51%) (*p* = 0.451), Spirochaetes (2.69% vs. 7.12%) (*p* = 0.185) and Cyanobacteria (0.14% vs. 0.40%) (*p* = 0.034) were lower (Figure 2B). At the genus level, gut microbial profiles differed between the two pig breeds. The top 35 genera across all samples are shown in Appendix A. In general, Lactobacillus, Clostridium, Terrisporobacter and Treponema were the four most abundant genera in the colons of both pig breeds. Furthermore, the abundances of the genera Ruminococcaceae_NK4A214_group (*p* = 0.040), *Parabacteroides* (*p* = 0.035), Chritensenellaaceae_R-7_group (*p* = 0.018) and *Ruminiclostridium* (*p* = 0.036) were higher in NX pigs compared with LW pigs, whereas the abundances of Prevotellaceae_NK3B31_group (*p* = 0.042), *Prevotella* (*p* = 0.027), *Subdoligranulum* (*p* = 0.023) and *Faecalibacterium* (*p* = 0.019) were lower (Figure 2C).

There was no significant difference in diversity (Shannon, Simpson, chao1 and ACE indices) between the two pig breeds (*p* > 0.05) (Appendix A). For beta diversity analysis, the colonic bacterial communities of the two pig breeds were compared using NMDS and ANOSIM based on Bray–Curtis distances. NMDS analysis revealed that the microbiota of the two pig breeds clustered separately along the principal coordinate axis (Figure 3A). The results of ANOSIM analysis of beta diversity indicated that there were significant differences in the gut microbiota structure between the two pig breeds (*R* = 0.382, *p* = 0.001) (Figure 3B), indicating that breed exerted a notable effect on microbial communities in the colon.

### 3.5. Functions of Colonic Microbiota

Due to significant differences in the composition of the gut microbiota, we performed functional analysis of the microorganisms using PICRUSt. Principal components analysis of functional profiles revealed that differences in the microbial functions of the two breeds (Figure 4) were consistent with the differences in composition described above. At the KEGG level, the gut microbiota of NX pigs involved more functions related to metabolic pathways, such as fatty acid metabolism, xenobiotics biodegradation and metabolism, flavonoid biosynthesis and disease, than that of LW pigs. In contrast, metabolism of carbohydrates, protein and biosynthesis of zeatin and carotenoid were enriched in LW pigs compared with NX pigs. The microbiota of LW pigs involved more functions related to energy metabolism, whereas the microbiota of NX pigs involved more functions associated with lipid metabolism and the immune system.

### 3.6. SCFAs Content in the Colon

As shown in Table 2, the total concentration of SCFAs, propionate and butyrate in the colon of NX pigs was lower than that of LW pigs (*p* < 0.05), and there was a trend toward lower concentrations of valerate in the colon of NX pigs (*p* = 0.094). There were no significant differences in branched chain fatty acids (BCFAs) between the two breeds (*p* > 0.05).

## 4. Discussion

Fats are the major form of energy storage in animals. Studies have shown that different pig breeds possess different growth potentials and fat deposition characteristics, both of which have profound effects on meat quality [26,27]. The NX pig, a well-known Chinese Indigenous fatty-type breed, exhibits early sexual maturity, high IMF content and better meat quality than other local pig breeds [28]. In contrast, LW pigs are a meat-producing breed known for their high growth rate and feed efficiency and their lean carcasses. In the current study, we found that NX pigs exhibited greater IMF content and backfat thickness but lower ADG and drip loss than LW pigs at 105 days of age. These results are in accordance with previously reported results [29]. However, the mechanisms underlying variations in meat quality remain unknown.

The colon is the main site of microbial fermentation [30], and the core flora in the intestine directly affects the function of the gut [31]. Previous studies have demonstrated that the gut microbiome differs from that of the large intestine in different breeds of pigs [32,33]. It has also been shown that gut microbes not only provide energy for life sustaining activities but are also involved in regulating body lipid storage [34]. Pigs fed the same diet have different gut microbiota profiles involving different microbial species [35], whereas in pig fecal samples, Firmicutes and Bacteroidetes predominate at the phylum level [36]. The present study demonstrated for the first time the differences in the intestinal microbiota of NX and LW pigs. We found that Firmicutes and Bacteroidetes were the dominant bacteria in the colonic contents of both pig breeds, but the Firmicutes/Bacteroidetes ratio of NX pigs (2.55) was slightly higher than that of LW pigs (2.23). Higher Firmicutes/Bacteroidetes ratios are associated with greater energy absorption and accumulation [31]. It has been shown that the abundance of Firmicutes is higher in the intestine of obese pigs, whereas the abundance of Bacteroidetes is lower [37]. Thus, it is possible that differences in Firmicutes and Bacteroidetes explained the higher IMF accumulation observed in NX pigs in this study.

In the current study, at the genus level, the relative abundances of genera including Ruminococcaceae_NK4A214_group, *Parabacteroides*, *Ruminantium*_group, Family_XIII_AD3011_group, Christensenellaceae_R-7_group and *Ruminiclostridium*_6 were higher in NX pigs than in LW pigs. Ruminococcaceae, Family_XIII and Christensenellaceae are members of the order Clostridiale, which is widely found in diverse gut communities [38] and is capable of degrading plant polysaccharides. They can also produce butyrate and acetate via the butyryl-coenzyme A (CoA):acetate CoA-transferase pathway [39]. Butyric acid is the main energy source of colonic mucosal epithelial cells, which maintain the structural integrity of the intestinal mucosa and promote the growth of the large intestine [40,41]. It is worth noting that butyrate exerts potent effects on a variety of colonic mucosal functions, such as the inhibition of inflammation and carcinogenesis and the reinforcement of various components of the colonic defense barrier [41,42]. It has been reported that a reduction in SCFAs produced by microbes in the gut leads to inflammation [43]. Furthermore, higher SCFAs concentrations have been observed in obese individuals [44]. In the present study, concentrations of SCFAs in the intestinal lumen of NX pigs were lower than those in LW pigs, although NX pigs possessed more abundant SCFAs-producing bacteria in the lumen of the colon than LW pigs. This discrepancy may be due to the fact that the colonocytes of NX pigs have a greater SCFAs-absorbing capacity than LW pig colonocytes [13]. In addition, analysis of the metabolic potential of the colonic microbiome using PICRUSt revealed that fatty acid biosynthesis pathways were enriched in NX pigs. Similar to a previous study that showed that fatty acid synthesis in human adipose tissue is linked to obesity and type 2 diabetes [45], the results of the current study indicated that the colonic microbiome may contribute to excess energy intake and increased body fat mass in NX pigs. Additionally, imbalances in the proportions of intestinal microbes contributed to the development of obesity by promoting the metabolism of energy from food, activating systemic inflammation and increasing lipid deposition [46,47]. Further studies on other aspects, such as transcriptomics and metagenomics, are necessary to explore the mechanisms underlying the different phenotypes of the two pig breeds.

## 5. Conclusions

The present study first compared the IMF content and the composition of the gut microbiota in the colon between LW and NX pigs. The findings revealed significant differences in body weight between LW and NX pigs at 105 days of age. NX pigs exhibited higher IMF content and backfat thickness than LW pigs. The bacterial communities in the colon of NX pigs were also more diverse than those in the colon of LW pigs. Furthermore, predictions of bacterial community functions indicated increased fatty acid biosynthesis in NX pigs than in LW pigs. The present results are useful to understand the differences between the microbiomes of NX and LW pigs and could be applied to improve meat quality.

## Figures and Tables

**Figure 1 animals-11-01862-f001:**
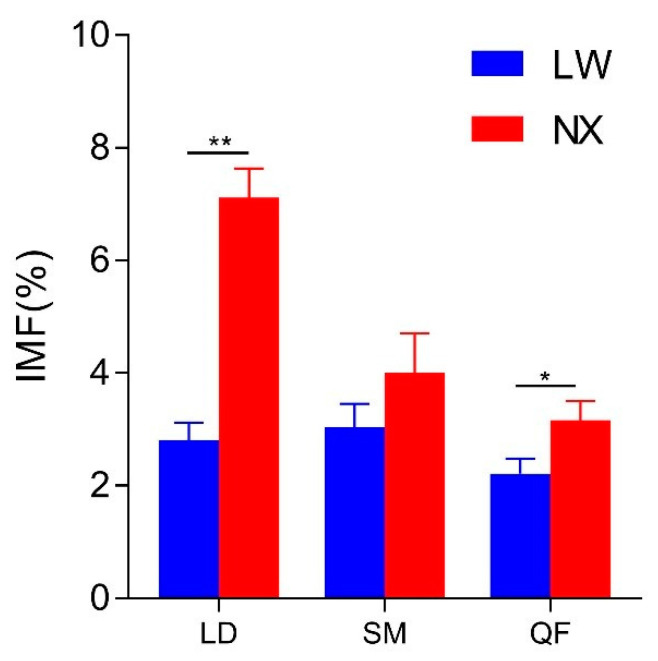
Intramuscular fat content of muscle between the two breeds. LD, longissimus dorsi muscle; SM, soleus muscle; QF, quadriceps femoris muscle. LW, Large White; NX, Ningxiang. ∗ indicates significant difference at *p* < 0.05, ∗∗ indicates significant difference at *p* < 0.01.

**Figure 2 animals-11-01862-f002:**
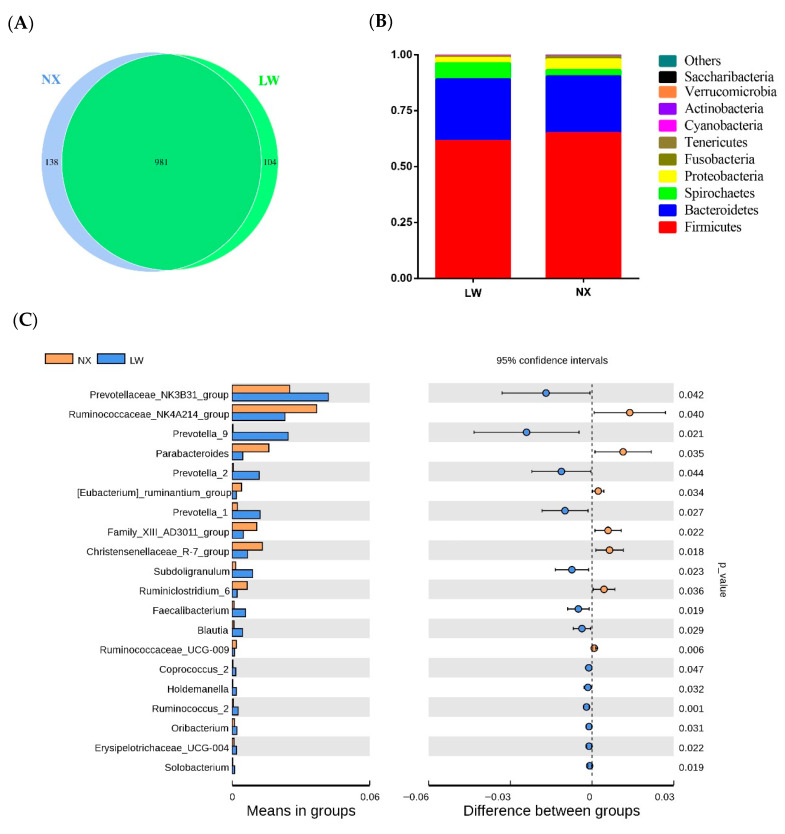
The analysis of colonic microbiota composition. (**A**) Venn diagram showing the coincidence of operational taxonomic units (OTUs) between the two pig breeds. (**B**) Phylum-level comparison of arithmetic mean plot with weighted Unifrac distance of β diversity. (**C**) Relative abundance of microbial genera (percentage) was significantly affected between the two pig breeds. LW, Large White; NX, Ningxiang.

**Figure 3 animals-11-01862-f003:**
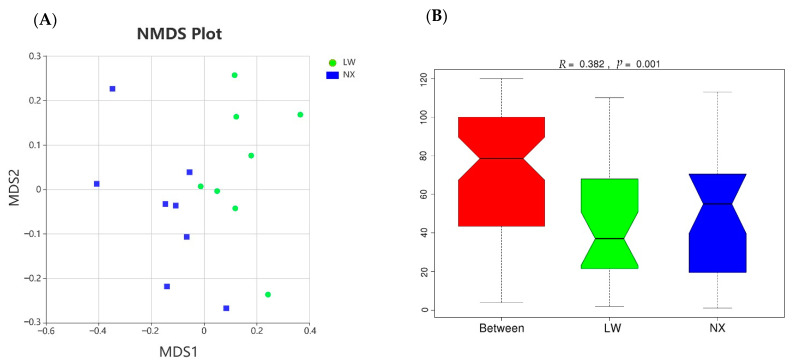
(**A**) Nonmetric multidimensional scaling (NMDS) plots of β diversity based on the relative abundance of OTUs in samples. (**B**) Anosim analysis of β diversity. LW, Large White; NX, Ningxiang.

**Figure 4 animals-11-01862-f004:**
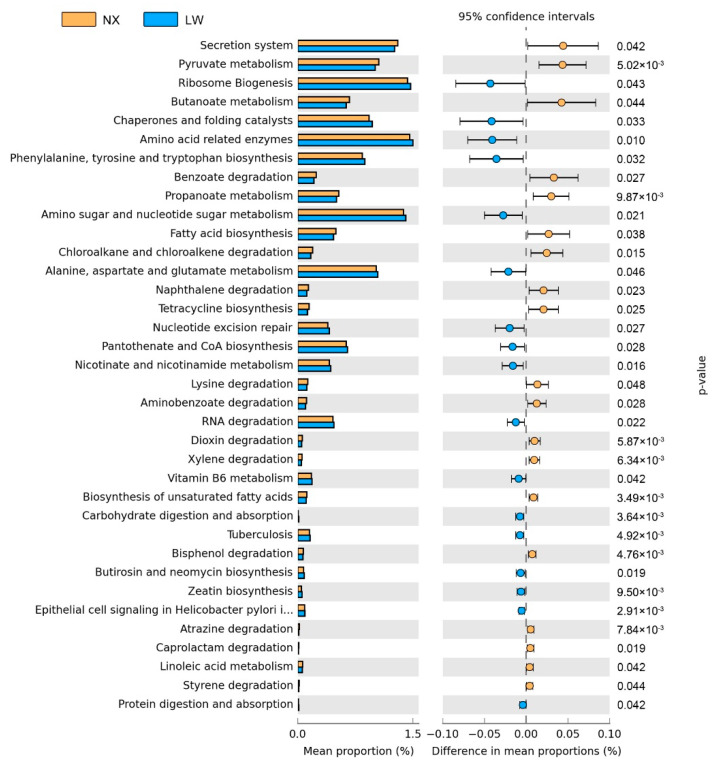
Prediction of changed the Kyoto encyclopedia of Genes and Genomes (KEGG) pathways using PICRUSt (plogenetic investigation of communities by reconstruction of unobserved states) analysis between the two pig breeds. LW, Large White; NX, Ningxiang.

**Table 1 animals-11-01862-t001:** Performances and carcass characteristics between Ningxiang pig and Large white pig.

Items ^1^	Genotype ^2^		SEM ^3^	*p*-Value
	LW	NX		
ADG (g/day)	842.26 ^a^	392.50 ^b^	11.84	0.025
ADFI (g/day)	1953 ^a^	1402 ^b^	35.03	0.037
F/G (g/g)	2.32 ^b^	3.60 ^a^	0.04	0.054
Final weight (kg)	99.34 ^a^	49.39 ^b^	1.35	0.018
Carcass weight (kg)	67.17 ^a^	30.98 ^b^	0.99	0.014
Dressing percentage (%)	67.57 ^a^	63.38 ^b^	0.63	0.048
Average backfat thickness (cm)	2.61 ^b^	4.54 ^a^	0.08	0.032
Drip loss (%)	5.21 ^a^	3.25 ^b^	0.06	0.029

^a,b^ Different superscript indicate statistical differences (*p* < 0.05). ^1^ ADG, average daily gain; ADFI, average daily feed intake; F/G, Feed/gain. ^2^ LW, Large White; NX, Ningxiang. ^3^ SEM, pooled standard error of means.

**Table 2 animals-11-01862-t002:** Concentrations of short-chain fatty acids (SCFAs) in the cecal digesta in Ningxiang pig and Large White pig.

Items ^1^	Genotype ^2^		SEM ^3^	*p*-Value
	NX	LW		
Acetate	9.19	12.65	0.820	0.107
Propionate	4.30 ^b^	7.02 ^a^	0.478	0.036
Butyrate	2.41 ^b^	4.88 ^a^	0.340	0.014
Isobutyrate	0.56	0.63	0.035	0.450
Valerate	0.67	0.95	0.064	0.094
Isovalerate	0.86	0.94	0.061	0.593
Total SCFAs	17.98 ^b^	27.07 ^a^	1.670	0.044
A/P ^4^	2.14	1.84	0.090	0.189
Total BCFAs	1.42	1.57	0.098	0.538

^a,b^ Different superscript indicate statistical differences (*p* < 0.05). ^1^ SCFAs, short-chain fatty acids; BCFAs, branched chain fatty acids. ^2^ LW, Large White; NX, Ningxiang. ^3^ SEM, pooled standard error of means. ^4^ A/P, ratio between acetate and propionate.

## Data Availability

All data are available from the corresponding authors on reasonable request.

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
