# Peer review of "Comparative Microbial Profiles of Colonic Digesta between Ningxiang Pig and Large White Pig"

_animals, 2021, doi:10.3390/ani11071862_

Round 1

Reviewer 1 Report

Dear Authors,

Your research on the comparison of the microbial profiles of the Ningxiang pigs and Large White pigs gut microbiome presented in the manuscript is very interesting and adds additional information to the knowledge related to the relationship between gut microbiota and adipogenesis.

My only remarks concern the revision of the editorial errors, such as: eliminating the extra space between lines 25 and 26, or unifying the font in Table 1.

Best regards.

Author Response

List of Responses

Dear Editors and Reviewers:

Thank you for your letter and for the reviewers,comments concerning our manuscript entitled “ Comparative microbial profiles of colonic digesta between Ningxiang pig and Large White pig ”(animals-1232895). Those comments are all valuable and very helpful for revising and improving our paper, as well as the important guiding significance to our researches. We have studied comments carefully and have made correction which we hope meet with approval. Revised portion are marked in red in the paper. The main corrections in the paper and the responds to the reviewer’s comments are as flowing:

Responds to the reviewer’s comments:

Reviewer: 1:

  1. Response to comment: My only remarks concern the revision of the editorial errors, such as: eliminating the extra space between lines 25 and 26, or unifying the font in Table 1.

Authors' response: We agree with the view. We have made correction according to the reviewer’s comments. Revised portion are marked in red in the paper.

Reviewer 2 Report

All the paper: The authors mentioned into the text must be changed by the reference number between square brackets as figure in the References’ paragraph.

Line 55: Please, change “intestinal flora” by “intestinal microbiota”.

Lines 100-101: I understand that feed intake was daily registered for each pig, but the description used in the text is equivocal.

Lines 101-103: Please, change “…and each pig was weighed at the beginning and end of the trial to calculate average daily gain (ADG), average daily feed intake (ADFI) and feed:gain ratio (F/G).” by “…and each pig was weighed at the beginning and the end of trial to calculate the average daily gain (ADG), the average daily feed intake (ADFI) and the feed:gain ratio (F/G).”

Lines 125-126: Please, change the paragraph to univocally remark that pig is the experimental unit for the total bacterial genomic DNA analysis. In my opinion the used phrase “…the colonic digesta of pigs in each group” is not totally clear. The phrase like “…the colonic digesta of each pig” is simpler and clearer.

Line 164: Please, change “PICRUSt can predict the function of 16S rRNA markers” by “PICRUSt can predict the principal functions of the corresponding 16S rRNA OTUs”.

Line 170: Please, change “dried at a low temperature” by “dried at low temperature”.

Lines 174-175: Please rewrite the sentence “The supernatant was taken and centrifugation was repeated, followed by mixing of the solution twice and adjustment of the final volume to 10 mL”.

Lines 236-240: When the P values are higher that 0.05 is not correct the introduce the concepts higher or lower.

Line 255: Please insert “the” before “two”.

Lines 274-275: Please rewrite and complete the sentence of Figure 4.

Line 286: Please, change “A/P, rate of acetate and propionate” by “A/P, ratio between acetate and propionate”.

Line 311 and 314: Please change “Bacteroides” by “Bacteroidetes”.

Lines 309-313: Please, clarify if the differences between breeds on the Firmicutes/Bacteroidetes ratios are significant. If no, it is not possible to use the concept “higher”.

Line 315-317: I believe that it would be essential to calculate the correlation degree (i.e. Pearson’s coefficient) between the obtained individual results for Firmicutes, Bacteroidetes and Firmicutes/Bacteroidetes ratio versus the IMF accumulation to include this sentence.

Lines 327-328: Please, change “promote growth of the large intestine” by “promote the growth of large intestine”.

Lines 358-359: Please, rewrite the sentence: “In addition, we found that SCFAs-producing bacteria may be a factor in obesity.”

Author Response

List of Responses

Dear Editors and Reviewers:

Thank you for your letter and for the reviewers,comments concerning our manuscript entitled “ Comparative microbial profiles of colonic digesta between Ningxiang pig and Large White pig ”(animals-1232895). Those comments are all valuable and very helpful for revising and improving our paper, as well as the important guiding significance to our researches. We have studied comments carefully and have made correction which we hope meet with approval. Revised portion are marked in red in the paper. The main corrections in the paper and the responds to the reviewer’s comments are as flowing:

Responds to the reviewer’s comments:

Reviewer: 2:

  1. Response to comment: The authors mentioned into the text must be changed by the reference number between square brackets as figure in the References’ paragraph.

Authors' response: We agree with the view. We have made correction according to the reviewer’s comments. Revised portion are marked in red in the paper.

  1. Response to comment: Line 55: Please, change “intestinal flora” by “intestinal microbiota”.

Authors' response: We agree with the view. We have made correction according to the reviewer’s comments. Revised portion are marked in red in the paper.

  1. Response to comment: Lines 100-101: I understand that feed intake was daily registered for each pig, but the description used in the text is equivocal.

Authors' response: We agree with the view. We have made correction according to the reviewer’s comments. Revised portion are marked in red in the paper.

  1. Response to comment: Lines 101-103: Please, change “…and each pig was weighed at the beginning and end of the trial to calculate average daily gain (ADG), average daily feed intake (ADFI) and feed:gain ratio (F/G).” by “…and each pig was weighed at the beginning and the end of trial to calculate the average daily gain (ADG), the average daily feed intake (ADFI) and the feed:gain ratio (F/G).”

Authors' response: We agree with the view. We have made correction according to the reviewer’s comments. Revised portion are marked in red in the paper.

  1. Response to comment: Lines 125-126: Please, change the paragraph to univocally remark that pig is the experimental unit for the total bacterial genomic DNA analysis. In my opinion the used phrase “…the colonic digesta of pigs in each group” is not totally clear. The phrase like “…the colonic digesta of each pig” is simpler and clearer.

Authors' response: We agree with the view. We have made correction according to the reviewer’s comments. Revised portion are marked in red in the paper.

  1. Response to comment: Line 164: Please, change “PICRUSt can predict the function of 16S rRNA markers” by “PICRUSt can predict the principal functions of the corresponding 16S rRNA OTUs”.

Authors' response: We agree with the view. We have made correction according to the reviewer’s comments. Revised portion are marked in red in the paper.

  1. Response to comment: Line 170: Please, change “dried at a low temperature” by “dried at low temperature”.

Authors' response: We agree with the view. We have made correction according to the reviewer’s comments. Revised portion are marked in red in the paper.

  1. Response to comment: Lines 174-175: Please rewrite the sentence “The supernatant was taken and centrifugation was repeated, followed by mixing of the solution twice and adjustment of the final volume to 10 mL”.

Authors' response: We agree with the view. We have made correction according to the reviewer’s comments. Revised portion are marked in red in the paper.

  1. Response to comment: Lines 236-240: When the P values are higher that 0.05 is not correct the introduce the concepts higher or lower.

Authors' response: We agree with the view. We have made correction according to the reviewer’s comments. Revised portion are marked in red in the paper.

  1. Response to comment: Line 255: Please insert “the” before “two”.

Authors' response: We agree with the view. We have made correction according to the reviewer’s comments. Revised portion are marked in red in the paper.

  1. Response to comment: Lines 274-275: Please rewrite and complete the sentence of Figure 4.

Authors' response: We agree with the view. We have made correction according to the reviewer’s comments. Revised portion are marked in red in the paper.

  1. Response to comment: Line 286: Please, change “A/P, rate of acetate and propionate” by “A/P, ratio between acetate and propionate”.

Authors' response: We agree with the view. We have made correction according to the reviewer’s comments. Revised portion are marked in red in the paper.

  1. Response to comment: Line 311 and 314: Please change “Bacteroides” by “Bacteroidetes”.

Authors' response: We agree with the view. We have made correction according to the reviewer’s comments. Revised portion are marked in red in the paper.

  1. Response to comment: Lines 309-313: Please, clarify if the differences between breeds on the Firmicutes/Bacteroidetes ratios are significant. If no, it is not possible to use the concept “higher”.

Authors' response: We agree with the view. We have made correction according to the reviewer’s comments. Revised portion are marked in red in the paper.

  1. Response to comment: Line 315-317: I believe that it would be essential to calculate the correlation degree (i.e. Pearson’s coefficient) between the obtained individual results for Firmicutes, Bacteroidetes and Firmicutes/Bacteroidetes ratio versus the IMF accumulation to include this sentence.

Authors' response: We agree with the view. The reviewer gave us a good suggestion , however, in fact, this sentence is our reference, I believe that we will intend to study this problem in future experiments.

  1. Response to comment: Lines 327-328: Please, change “promote growth of the large intestine” by “promote the growth of large intestine”.

Authors' response: We agree with the view. We have made correction according to the reviewer’s comments. Revised portion are marked in red in the paper.

  1. Response to comment: Lines 358-359: Please, rewrite the sentence: “In addition, we found that SCFAs-producing bacteria may be a factor in obesity.”

Authors' response: We agree with the view. We have made correction according to the reviewer’s comments. Revised portion are marked in red in the paper.

Reviewer 3 Report

The manuscript showed results of colonic digesta of different breeds of pigs fed  the same diet.

The manuscript is interesting but the authors should clarify and review some points.

Results - lines 228 – 233 – The variation obtained was not presented in the figures, such as “  Firmicutes were the most predominant, representing 47.36–74.27% of the bacterial population of both NX and LW”. If this parameter was related to eight pigs it is difficult to understand the significant diferences showed in the figure 2C.

Results lines 268 and 270 – The authors stated that the gut microbiota was  related to flavonoid and carotenoid BIOSYNTHESIS  in the gut . Please review

The conclusion should be review because the functions of the microbiota was based on the KEGG and they were not proved by this work. The conclusion about the SFCA producing bacteria  was not demonstrated by this work.

The description of the SFCA analysis failed. The solvent used for SFCA extraction and the conditions of th GC analysis were not reported.  It is not clear if the digesta of eight pigs per breed was evaluated. This information was ommitted for IMF and Drip loss.

The results of drip loss were not discussed.

Material and methods lines 121 – 123 – This paragraph should de reviewed. “The IMF ... after extraction??

Author Response

List of Responses

Dear Editors and Reviewers:

Thank you for your letter and for the reviewers,comments concerning our manuscript entitled “ Comparative microbial profiles of colonic digesta between Ningxiang pig and Large White pig ”(animals-1232895). Those comments are all valuable and very helpful for revising and improving our paper, as well as the important guiding significance to our researches. We have studied comments carefully and have made correction which we hope meet with approval. Revised portion are marked in red in the paper. The main corrections in the paper and the responds to the reviewer’s comments are as flowing:

Responds to the reviewer’s comments:

Reviewer:3:

  1. Response to comment: Results - lines 228 – 233 – The variation obtained was not presented in the figures, such as “  Firmicutes were the most predominant, representing 47.36–74.27% of the bacterial population of both NX and LW”. If this parameter was related to eight pigs it is difficult to understand the significant diferences showed in the figure 2C.

Authors' response: We agree with the view. In fact, “Firmicutes were the most predominant, representing 47.36–74.27% of the bacterial population of both NX and LW”, we test the composition of two kinds of pigs, however, the figure 2C is “ relative abundance of microbial genera (percentage) was significantly affected between the two pig breeds”. There is no contradiction between them.

  1. Response to comment: Results lines 268 and 270 – The authors stated that the gut microbiota was  related to flavonoid and carotenoid BIOSYNTHESIS  in the gut . Please review

Authors' response: We agree with the view. In fact, in this study, according to  Figure 4. Prediction of changed KEGG pathways using PICRUSt analysis between the two pig breeds. Revised portion are marked in red in the paper.

  1. Response to comment: The conclusion should be review because the functions of the microbiota was based on the KEGG and they were not proved by this work. The conclusion about the SFCA producing bacteria  was not demonstrated by this work.

Authors' response: We agree with the view. We have made correction according to the reviewer’s comments. Revised portion are marked in red in the paper.

  1. Response to comment: The description of the SFCA analysis failed. The solvent used for SFCA extraction and the conditions of th GC analysis were not reported.  It is not clear if the digesta of eight pigs per breed was evaluated. This information was ommitted for IMF and Drip loss.

Authors' response: We agree with the view. We have made correction according to the reviewer’s comments. Revised portion are marked in red in the paper.

  1. Response to comment: The results of drip loss were not discussed.

Authors' response: We agree with the view. We have discussed it in the discussion section. Revised portion are marked in red in the paper.

  1. Response to comment: Material and methods lines 121 – 123 – This paragraph should de reviewed. “The IMF ... after extraction??

Authors' response: We agree with the view. We have made correction according to the reviewer’s comments. Revised portion are marked in red in the paper.